# What polarizes citizens? An explorative analysis of 817 attitudinal items from a non-random online panel in Germany

**Céline Teney**[1]*, **Giuseppe Pietrantuono**[1,2], **Tobias Wolfram**[3,4]

**1** Institute of Sociology, Freie Universität Berlin, Berlin, Germany, **2** Immigration Policy Lab, ETH Zurich, Zurich, Switzerland, **3** Faculty of Sociology, Bielefeld University, Bielefeld, Germany, **4** Civey Research Institute, Berlin, Germany

* celine.teney@fu-berlin.de

## Abstract

Various studies point to the lack of evidence of distributive opinion polarization in Europe. As most studies analyse the same item batteries from international social surveys, this lack of polarization might be due to an item's issue (e.g., the nature or substance of an item) or item formulation characteristics used to measure polarization. Based on a unique sample of 817 political attitudinal items asked in 2022 by respondents of a non-random online panel in Germany, we empirically assess the item characteristics most likely to lead to distributive opinion polarization–measured with the Van der Eijk agreement index. Our results show that only 20% of the items in our sample have some–at most moderate–level of opinion polarization. Moreover, an item's salience in the news media before the survey data collection, whether an item measures attitudes toward individual financial and non-financial costs, and the implicit level of knowledge required to answer an item (level of technicality) are significantly associated with higher opinion polarization. By contrast, items measuring a cultural issue (such as issues on gender, LGTBQI+, and ethnic minorities) and items with a high level of abstraction are significantly associated with a lower level of polarization. Our study highlights the importance of reflecting on the potential influence of an item's issue and item formulation characteristics on the empirical assessment of distributive opinion polarization.

## Introduction

Opinion polarization has become a prevalent research and mediatic topic in Western Europe over recent years, mainly due to its scientific and societal relevance. Indeed, while multiple and cross-cutting opinion polarization might be conducive to social order in pluralist societies, opinion polarization combined with issue alignment can lead to political conflicts and thus threaten social cohesion and social order [1, 2].

Launched in 1996 with the landmark work of DiMaggio et al. [2], the debate about the extent of opinion polarization within the U.S. population has received increasing attention from the social sciences community (see [3] for an overview). The U.S. offer fruitful soil to this debate as it constitutes a unique case in which the socio-political context is particularly

**Data Availability Statement:** The dataset is available under: Teney, C., Pietrantuono, G., & Wolfram, T. (2023, September 29). Dataset of

Civey Items on Polarisation. https://doi.org/10.17605/OSF.IO/BFCVH.

**Funding:** This paper is part of a research project on attitudinal polarization funded by the German Research Foundation (DFG). The funders had no role in study design, data collection and analysis, decision to publish, or preparation of the manuscript.

**Competing interests:** No authors have competing interests.

propitious for opinion polarization–given the two-party system, the siloed and partisan medial landscape, and the partisan residential segregation. Largely influenced by the U.S. opinion polarization debate, European social scientists have been increasingly investigating the extent of opinion polarization in the European context. A consistent finding from these studies is the overall lack of evidence of (distributive) opinion polarization.

Distributive opinion polarization is characterised by a situation in which citizens position themselves on the opposite edges of an attitudinal divide [3, 4]. Empirically, one would speak of distributive opinion polarization if attitudinal items show a bimodal distribution or are closed to such a bimodal distribution [2, 3]. Cross-national European studies on both cultural issues such as immigration, EU, gender and LGTBQI+ and economic issues [e.g., 5–7] highlighted the lack of evidence of distributive opinion polarization. Such diagnostics also hold for studies on the German case [8]. Even long-trend studies covering several decades of survey data conclude that distributive opinion polarization on cultural issues has not increased over time ([9] for a cross-national analysis [10, 11], for the German case).

We argue that this lack of evidence of distributive opinion polarization might be due to the characteristics of items used to study polarization: Generally, previous studies based their analysis on items available in secondary national and international social sciences surveys such as the *European Social Survey* or the *International Social Survey Program*. Attitudinal items typically used in such large-scale social sciences surveys tend to measure attitudes toward very general issues or broad principles, such as whether immigration enriches cultural life or whether the state should reduce social inequality. Furthermore, most studies used the same item batteries to assess polarization, as most international and national social surveys comprise items with a very similar formulation. However, the formulation of an item affects the polarizing power of survey items (see, for instance, [10]). Hence, this debate would greatly benefit from reflecting on the formulation characteristics of the typical survey items analysed and the potential effects of survey measurement on distributive opinion polarization.

With this study, we seek to understand the item characteristics that most likely lead to distributive opinion polarization, by analyzing a unique set of 817 political attitudinal items, which were gathered from a non-random online panel in Germany during 2022. We investigate the role of item characteristics in explaining the modality of the item response distribution. We take two characteristics into account: (a) the item's issue (e.g., the nature or substance of an item), and (b) the item's formulation. Arguably, we focus in this study on the simplest form of opinion polarization measured with the modality of an item's distribution among the overall sample of survey respondents, leaving aside more complex forms of polarization, such as group-based polarization [12] or affective polarization [13]. This focus on a single and simple form of opinion polarization is necessary to launch a scientific debate targeting the role of item formulation and the issue of items in opinion polarization. By drawing on the literature in psychology, sociology, and political sciences, we derive hypotheses on six item characteristics that we expect to influence distributive opinion polarization. The first two relate to the item's issue, the last four to the way items are formulated: (i) the extent to which an item issue was salient in the news media before the survey data collection, (ii) the extent to which an item tackles a cultural issue, (iii) the extent to which an item involves individual costs or benefits, (iv) the extent to which an item targets a minorty group, (v) the level of abstraction of an item, and (vi) the level of technicality of an item.

Our results show that only 20% of our sample of items has some–at most moderate–level of opinion polarization. Moreover, whether an item tackles financial and non-financial costs, its salience in the news media, and the implicit level of knowledge required to answer an item (level of technicality) are significantly associated with higher opinion polarization. By contrast, items measuring a cultural issue and items with a high level of abstraction are significantly

associated with a lower level of polarization. Following the development of our hypotheses, we present our sample, the operationalization of distributive opinion polarization and our five item characteristics before interpreting our statistical analysis of item characteristics on distributive opinion polarization.

## Item characteristics and their impact on distributive polarization

While survey measurement research provides many general recommendations about survey item formulation [e.g., 14, 15], it does not specify the conditions under which a survey item is particularly likely to polarize respondents. We, therefore, draw on the literature in psychology, sociology, and political sciences to define two hypotheses on item issue characteristics and four hypotheses on item formulation characteristics and their role in distributive opinion polarization. In this section, we present our hypotheses on these item characteristics.

### Item issue salience in the news media

Our first hypothesis on item issue characteristics refers to the role of media issue salience in distributive opinion polarization. From the psychological theory of directly motivated reasoning [16], we can hypothesize that mere exposure to information on an issue through the media is likely to lead to distributive opinion polarization. Directly motivated reasoning refers to the (unconscious) strategy of people to seek out information that reinforces their preferences (i.e., confirmation bias), denigrate attitudinal incongruent arguments (i.e., disconfirmation bias), and evaluate information supporting their prior attitudes as stronger and more compelling than counter attitudinal information (i.e., prior attitude effect) [17] (p. 757). Directional motivational reasoning implies that processing additional information on an issue is likely to sharpen citizens´ prior beliefs and attitudes on the particular issue, which in turn increases attitudinal polarization [17]. Empirical studies have indeed shown that directly motivated reasoning leads citizens to endorse stronger opinions (i.e., be more polarized) on an issue after having been exposed to new information on this issue. This effect appears in particular among those who have strong prior opinion on the respective issue and those who are more politically knowledgeable, as the former have affective links to the issue and the latter possess more ammunition to counter information disconfirming their prior beliefs [17, 18].

The (unconscious) activation of directly motivated reasoning is independent of the content of the information to which one is exposed (i.e., whether the information content confirms or disconfirms prior beliefs) [17]. Thus, the mere exposure to media news on an issue is likely to induce directly motivated reasoning among citizens (in particular, those with strong prior beliefs and those more politically knowledgeable) who would then hold more polarized opinions on the respective issue. Indeed, Wojcieszak et al. [18] showed that citizens in the Netherlands who were both fervent supporters and opponents of the EU held more polarized opinions after being exposed to media news about the EU. Therefore, we expect items with a high issue salience in the media to be more polarizing. By media issue salience, we refer to the relative coverage the news media allocates to a given issue [19].

*H1*: *The higher the media salience of an item's issue, the more polarizing the item will be.*

### Items on cultural issues

Our second hypothesis on item issue characteristics focuses on the extent to which items measuring cultural issues polarize more than items measuring other issue domains. Literature on political cleavage and value change suggests that citizens are increasingly divided on cultural-

related issues such as gender, LGTBQI+, or immigration [e.g., 20–23]. Most prominently, Norris and Inglehart [24] develop this thesis in their book "The cultural backlash". Accordingly, the long-term societal value shift toward cultural liberalization (or "postmaterialism" in Inglehart´s terminology) that started in the aftermath of WWII in advanced democracies has triggered an authoritarian reflex among those who feel most threatened by these changes as they fear losing their majority status. Those who feel threatened by this societal value shift are likely to react by endorsing more authoritarian attitudes on cultural issues, such as the rejection of diverse lifestyles of groups that are perceived as violating conventional norms and traditional customs, including anti-LGTBQI+ and anti-immigrant attitudes or opposition to gender emancipatory roles and norms [24]. According to the cultural backlash theory, the value shift towards cultural liberalization is expected to have deepened opinion polarization on cultural issues in many advanced democracies.

Empirical studies on the evolution of a cultural backlash in the European context were inconclusive: opinion divide on cultural issues along generational or educational lines (the two key socio-demographic characteristics mentioned by [24]) has not increased in Western Europe [9, 25]. Nevertheless, and more importantly for our purpose, the political cleavage literature highlights that cultural-related issues have become more salient and conflictual than other issues -such as socio-economic issues- among the Western European population [e.g., 20, 23]. By extension, whether an item measures a cultural issue could affect its polarizing level. We, therefore, hypothesise that:

*H2*: *Items on cultural issues are more likely to polarize than items on non-cultural issues.*

## Item targeting individual costs and benefits

Turning now to item formulation characteristics, we introduce the distinction between items measuring individual costs and items measuring individual benefits. Prospect theory [26, 27] provides a theoretical framework for explaining why items implying individual costs might be more polarizing than items implying individual benefits. According to Kahneman and Tversky [27], people perceive the outcome of a decision in terms of gains and losses defined with a reference point (usually the current state) rather than in terms of the final stage. Indeed, "the Humans described by prospect theory are guided by the immediate emotional impact of gains and losses, not by long-term prospects of wealth and global utility" [26]. This tendency helps us understand how respondents tend to answer attitudinal items on implementing a policy or political measure: respondents tend to evaluate the kind of *changes* the policy implementation would imply in terms of losses or gains to them personally.

Moreover, Kahneman and Tversky demonstrated that people put more weight on losses than on gains: "The aggravation that one experiences in losing a sum of money appears to be greater than the pleasure associated with gaining the same amount" [27]. This cognitive mechanism makes individuals loss aversive in their behaviour by preferring the avoidance of losses over the acquisition of equivalent gains. In addition and more critical to our purpose, it implies that individuals respond more to losses than gains ([26], see also [28] for empirical evidence). Pierson [29] pointed to the prospect theory's relevance to understand the imbalance between citizens' reactions to social policy cuts and welfare expansion. Moreover, this tendency of reacting more strongly to losses than to gains might lead to a more extensive opinion polarization toward policies or a political matter implying individual costs (such as the implementation of a new tax) than toward policies or a political matter implying individual benefits (such as the allocation of a financial bonus). Indeed, a stronger reaction to attitudinal items on

individual costs would concretely manifest in a larger tendency to use the extreme answer categories when answering attitudinal items measuring individual costs. This, in turn, leads to larger opinion polarization at the aggregate level.

Furthermore, Kahneman and Tversky [27] expect this loss aversion mechanism to apply to decisions about both financial and non-financial attributes. Accordingly, respondents are more likely to react more strongly to items measuring attitudes towards financial and non-financial individual costs than to items measuring financial and non-financial individual benefits. An example of a non-financial individual cost item from our sample concerns the implementation of a highway speed limit. An example of a non-financial individual benefit item from our sample is lifting the COVID-test obligation to attend schools during the pandemic. In sum, prospect theory enables us to derive the following hypothesis:

H3: *Items measuring attitudes towards financial and non-financial individual costs polarize to a larger extent than items measuring attitudes toward financial and non-financial individual benefits.*

## Items targeting minorities

Another item formulation characteristic that might be related to distributive polarization is the distinction between minority and non-minority targets. The literature on identity politics can help us use this distinction to draw a further hypothesis on the potential polarizing power of minority targets. Identity politics refers to policies aiming to improve the position and status or strengthen the societal recognition of individuals because of their membership in an underrepresented group. Identity politics involve deliberate group boundary-making that implies excluding the "Other". Recently, several scholars drew attention to some societal drawbacks of identity politics. For instance, Fukuyama [30] pointed to the risk of identity politics turning into excessive political particularism: identity politics implies focusing on ever smaller underrepresented groups while neglecting to build inclusive solidarity around large collectivities. In a similar vein, Lilla [31] argues that identity politics is divisive as those who are not targeted by identity politics feel excluded (such as the white male working class) (see also [32]).

Some European scholars also discussed the potential drawbacks of identity politics as it can trigger fragmentation and hamper social solidarity or a shared sense of collective purpose [e.g., 33]. These criticisms shed light on the risk of societal fragmentation and lack of inclusive solidarity around large collectives resulting from a (policy) focus on underrepresented or minority groups. Furthermore, despite their objective of belonging to a privileged group, members of the majority can feel excluded if they perceive their group as less valued or less recognized [34]. Policies targeting exclusively minority groups can enhance such feelings of exclusion: policies targeting minorities while excluding the majority group are significantly more divisive than all-inclusive policies [35, 36]. By extension, we expect items targeting a minority group to be more polarizing than other items. Items targeting a minority group are more likely to be perceived as securing or implementing particularistic interests, leading to a more affective and stronger response and, thus, polarization. An item example from our sample targeting a minority is "Are you in favour of maintaining dual citizenship in Germany?". This leads us to formulate the following hypothesis:

H4: *Items targeting a minority group are more likely to polarize.*

## Concrete vs. abstract item formulation

Our third hypothesis on item formulation characteristics refers to the item level of abstraction. The literature suggests that people often respond more strongly to objects or events that are more immediate in both time and location and are actual rather than hypothetical (see, for instance [37]).

First, a study by Dochow-Sondershaus and Teney [10] points to the fact that immigration-related attitudinal items measuring particular policies–such as restriction on immigration flow–were more likely to be divisive across occupation classes in Germany. By contrast, items measuring broader principles on immigrants, such as the extent to which immigration is good or bad for the economy–a typical anti-immigrant survey item included in various international social surveys–are more likely to provoke consensual opinion among Germans regardless of their occupational classes. Thus, the level of abstraction in the item formulation seems to play a role in polarizing respondents.

This finding can be interpreted in light of the construal level theory. According to the review by Trope and Liberman [37], "a high level of construal can be defined as a relatively abstract, coherent and superordinate mental representation of an object ". The extent to which an object has a high or low level of construal representation influences the intensity of affective response to the object [37]. Affective or stronger reactions to objects with a low construal level in answering survey items would manifest in selecting more extreme answer categories. Thus, they would result in opinion dissent or polarization at the aggregate level. An example from our items sample with a low construal level is "In your opinion, should the German government decide to supply battle tanks to Ukraine?". An example from our items sample from the same issue domain with a high construal level is "Do you have a more positive opinion on the German military since the beginning of the war in Ukraine?"

Second, having psychological distance from an object reduces the tendency to respond to the object by drawing on one's preconceptions while increasing openness to divergent viewpoints [38]. Manipulating objects to increase their level of construal representation has been shown to reduce opinion divergence between conservatives and liberals in the US: increasing the level of construal representation enhances "the capacity of both liberals and conservatives to consider each other's positions and the potential to find common ground between them" [39]. In addition, high-level construal objects are more likely to require the activation of core values and ideals, such as fairness and justice, that are likely to be shared within important groups [40].

Activating such broader societal values, in turn, is expected to reduce partisan polarization insofar as everyone -regardless of partisan preferences- cares to some extent about fairness and justice [41]. In contrast, low-level construal objects include peripheral and contextual aspects, which enable context-dependent evaluations based on unique details of the present situation [40]. Attitudes towards low-level construal objects are therefore less likely to be convergent, as respondents build their opinion by retrieving context-dependent information and unique details of the object that are likely to be perceived and evaluated differently across respondents. Applying the construal level theory to items on policy-related issues implies that respondents overall agree on the goals of most policies (i.e., reducing social inequality or poverty) but disagree on the instruments (such as concrete policies) to reach these goals.

In sum, objects (in our case, survey items) with a high level of construal representation are less likely to attract affective or strong responses and to be based on preconceptions. At the same time, they are more likely to activate broader commonly shared values and openness to other viewpoints. This, in turn, can explain why high-level construal objects seem less polarizing than low-level construal objects. In other words, according to the construal level theory, survey items formulated in a more abstract way (high level of construal representation) are less

likely to polarize than survey items formulated in a more concrete way (low level of construal representation). This leads us to formulate our fifth hypothesis:

*H5*: *The more concrete the formulation of a survey item, the more it polarizes respondents.*

## Item level of technicality

Our last hypothesis focuses on the role of item level of technicality in distributive polarization. For this, we draw on both the work of Carmines and Stimson [42] on the distinction between easy and hard political issues and on the work of Converse [43] on voters' level of political sophistication.

According to Carmines and Stimson [42], the distinction between easy and hard political issues is essential for a better understanding of issue voting. Easy issues imply so-called *gut responses*, "Because gut responses require no conceptual sophistication, they should be distributed reasonably evenly in the voting population" [42] (p. 49). Thus, all citizens–regardless of their level of political interest, political knowledge, or level of education–possess the ability to express their own opinion when answering such easy issue items. By contrast, the discriminatory power of hard issue items is likely to be higher among citizens who are more politically interested, informed, and involved than among citizens less interested, informed, and involved. In other words, it will be mostly citizens with a high level of political sophistication and political interest who will be able to give a valid answer expressing their own opinion on a hard issue. By contrast, citizens lacking political knowledge and involvement are more likely to give random answers to hard-issue items. Thus, item measurement error and consequently item variation are likely to be larger on hard issues than on easy ones. We would therefore expect more distributive polarization on easy issues than on hard issues as positions on easy issues are measured with more accuracy.

Converse [43] comes to a similar conclusion in his seminal work on the nature of belief systems in the mass public when analyzing the implications of varying levels of political sophistication among voters. According to him, respondents with a lower level of political sophistication show political positions that are more random and less structured than respondents with a higher level of sophistication. He argues that a lack of political information and contextual grasp among citizens leads to the inability to relate one's ideology and own beliefs to a particular political issue. From an item perspective, items requiring a high amount of political and contextual information tend to show more variation and randomness in their answers than items not requiring such information [43].

Carmines and Stimson [42] conceptualized hard and easy issues by building on three complementary dimensions: level of technicality, measurement of policy ends and means, and length of the salience of an issue on the political agenda. We already drew hypotheses on two out of the three dimensions of easy and hard issues (i.e., hypothesis on level of abstraction -including the distinction between policy means and policy ends- and hypothesis on media issue salience). We, therefore, restrict this last hypothesis to the level of technicality of an issue, which enables us to construct a unidimensional indicator for measuring the easiness of an issue. Accordingly, technical issues require knowledge of important factual assumptions [42]. Hence, our last hypothesis is that the higher the level of technicality, the lower the level of polarization of an item:

H6: Items with a low level of technicality polarize more than items with a high level of technicality.

## Materials and methods

### Sample

The German public opinion research institute Civey collected the data used for this study. Civey conducts online surveys through a network of partner websites, including prominent German online newspapers. The Civey panel comprises around one million German citizens who registered with their email addresses and created profiles with personal information such as age, gender, and zip code. Additional respondent details are collected over time. Civey utilizes nonprobability sampling and takes various measures to guard against self-selection biases, described in detail by Richter, Wolfram, and Weber [44]. All items underwent the standard Civey weighting procedure, combining quota sampling and poststratification across demographic and political variables. This procedure includes factors such as sex, age, voting behavior, regional density, and purchasing power of the zip region.

The original sample is composed of 1491 unique items that were asked to on average 5000 respondents between January 1, 2022, and October 14, 2022, date on which we retrieved the dataset. Ethical approval was not necessary as our analysis is based on the reuse of a dataset collected by the Civey research institute among its panel members who consented to participate. We had only access to meta information on each survey item (e.g., date of survey, item distribution, item formulation) without any access to respondents´ individual answer to the survey items. All items were asked with a five-point Likert scale. We restricted the sample to attitudinal items by using the following definition of attitudes: "attitude represents a summary evaluation of a psychological object captured in such attribute dimensions as good-bad, harmful-beneficial, pleasant-unpleasant, and likeable-dislikable" [45]. We, therefore, excluded items asking about knowledge or beliefs.

Furthermore, we restricted the sample to items measuring attitudes toward a political matter. We used Almond and Verba´s definition of the political system to distinguish political from non-political attitudinal items: "In treating the component parts of the political system we distinguish [. . .] three broad classes of objects: (i) specific *roles* or *structures*, such as legislative bodies, executives, or bureaucracies; (ii) *incumbents* of roles, such as particular monarchs, legislators, and administrators, and (iii) particular public *policies*, *decisions*, or *enforcements* of decisions. These structures, incumbents, and decisions may in turn be classified broadly by whether they are involved either in the political or "input" process or in the administrative or "output" process" [46]. Furthermore, we excluded items beyond the context of Germany, the EU, Europe, or NATO. Lastly, we deleted all items asking about preferences toward one particular political actor without regard for the political actor´s behaviour or discourse (such as "How satisfied are you with Chancellor Olaf Scholz?") as these items only measure party or personality preferences. Our restricted sample comprises 817 items that meet all mentioned selection criteria.

### Operationalisation of the dependent variable

The agreement index proposed by Van der Eijk [47] constitutes a suitable measurement of distributive opinion polarization for ordinal variables, such as attitudinal items with Likert-scale answer categories. It has been widely used in previous studies on distributive opinion polarization [e.g., 11, 48]. The agreement index is computed with the proportion of respondents in contiguous answer categories. Applied to the example of a 5-point Likert scale, high agreement implies that most cases are found in contiguous answer categories, for example, the first and second ones. High disagreement, by contrast, means that observations fall into non-contiguous answer categories, for instance, 30% in the first answer category, 30% in the third answer

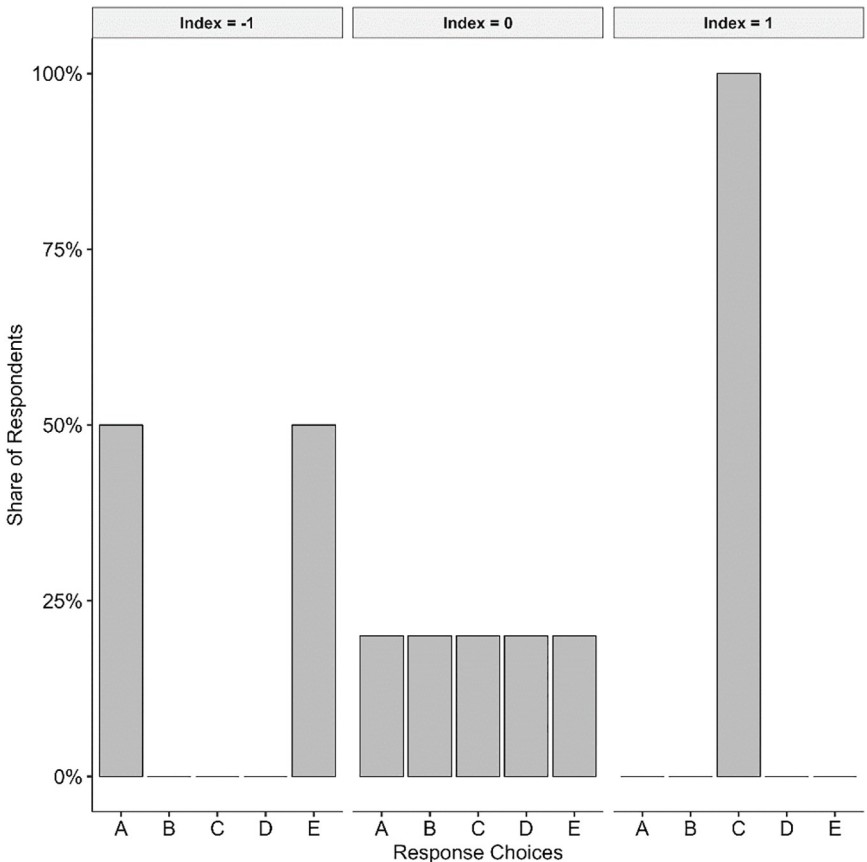

**Fig 1. Van der Eijk index.**

category and 40% in the fifth answer category. S1 Appendix provides a more detailed description of its computation. The agreement index ranges from −1 to 1 and is presented in Graph 1. A value of −1 denotes complete polarization- (perfect bimodal distribution) where 50% of all respondents fall into the first and the other 50% into the last answer category. A value of 1 refers to complete consensus, where all observations are to be found in a single category (perfect unimodality). An index value of 0 refers to a uniform distribution, where each answer category comprises an equal number of responses (see Fig 1).

## Operationalisation of independent variables

We developed a comprehensive coding scheme to build the independent variables. Besides the continuous salience variables, all other independent variables are categorical with 2 to 3 categories. As the semiotic nuance between some sampled items is particularly granular, our coding scheme provides various examples for each variable category to ensure coding reproducibility. Furthermore, we computed an intercoder reliability test for each categorical variable based on a random sample of 100 items.

**Salience variable.**   To measure the *salience* of an item and the item's issue domain, we researched how often keywords related to the issue domain of an item are reported in newspapers. We relied on the "Genios" (available at www.genios.de) database. The database provides archive access to newspaper and magazine articles worldwide. This includes over 400 titles

from local, national, and international newspapers and around 1,000 trade and consumer magazines. For our purpose, we limited the search to German newspapers. We coded a measure capturing the frequency of mentions of the item's issue domain seven days before the day the item was included in the Civey online panel and logarithmized it.

**Issue domains and cultural issue variable.** We first coded all items into 20 issue domains by following Hutter and Kriesi's [49]coding scheme and including the issue domain "political trust". We then regrouped items belonging to the issue domains "ethnic diversity and immigration", "gender", "LGTBQI+", and "Cultural liberalism: other". We built a dummy *cultural issue* variable with the value 1 for items belonging to a cultural issue domain and 0 for the other items.

**Costs and benefits variables.** We coded two dummy variables indicating whether an issue implies financial or non-financial costs or benefits for individuals:

The variable measuring *costs* is coded as one whenever an issue suggests generating costs or externalities for society. This variable comprises financial charges, such as tax-related items and non-financial costs, such as bans. Exemplary items measuring costs are: "Would you be willing to pay more for electricity from renewable sources?" or "Should the public display of imperial war flags be banned nationwide?"

In analogy, we coded the variable measuring *benefits* as one if the item broadly implies financial (e.g., "Should purchasing cars with low CO2 emissions be given a tax incentive?") or non-financial benefits for (particular) individuals (e.g., "Should so-called "whistle-blowers" be better protected from prosecution in the future?"). In our study, we operationalize "costs" and "benefits" by coding the exact wording used in the items. Obviously, most items formulating a financial or non-financial cost (for instance, the implementation of a new tax) could be formulated the other way around by stressing the financial or non-financial benefits of such a policy (for instance, increasing the financial budget that would result from implementing a new tax). Moreover, items formulated by highlighting a cost can be interpreted by respondents as introducing a benefit. However, and for the sake of consistency, we focused our coding exclusively on the wording used in the item formulation that tap at (financial and non-financial) costs or benefits.

**Minority target variable.** We measured whether an item targets a minority group with a binary variable. We understand minorities broadly: whenever an issue targets a clearly defined group, not the whole population, the variable *minority* is coded as 1. These two items are exemplary for the variable: "Should gay couples have the same adoption rights as heterosexual couples?"; and "Would you support a general Corona vaccination requirement starting at age 60?"

**Level of abstraction.** The *abstraction* variable is a three-categorical variable measuring an item's level of abstraction and concreteness. Category 1 contains very concrete items on a particular policy or sub-policy with a narrow scope or application to specific or rare situations or temporary. Examples in this category are: "Should the federal government recognize Jerusalem as Israel's capital?" or "How would you evaluate the abolition of the obligation to wear masks on airplanes to and from Germany starting in the fall?". The second category comprises items on policies with a middle-range scope, applicable to a broad range of situations or often. Examples are "In your opinion, should all COVID-measures be terminated in the spring of 2022 if the infection situation is stable?" or "Do you think the fuel rebate introduced by the federal government to relieve the population in view of the high fuel prices makes sense?". The third category comprises abstract items on general principles or on an entire policy field not reduced to a time frame or on the overall level of satisfaction with a political institution. Examples for the third category include: "Does Islam belong to Germany?" or "How much trust do you have in the German constitutional state?"

**Level of technicality.** We measured the level of *technicality* with three categories: Items in category 1 require from respondents a high level of political sophistication to give a valid answer. Examples for this category are: "How satisfied are you with the work of the Federal Minister of Construction, Klara Geywitz?" or "How do you evaluate the fact that the federal government wants to include stocks more in pension planning in the future". The second category comprises items with an intermediated required level of sophistication. Examples are "Should Turkey remain a NATO member?" or "How much confidence do you have in the German rule of law?". The third category is composed of items requiring the lowest level of technicality. Examples for the third category include: "Would you describe yourself as a pacifist?" or "Does Islam belong to Germany?"

**Control variables.** The time frame covering the item sample coincided with two major events that dominated public debate: the COVID pandemic and the war in Ukraine. This naturally resulted in a higher representation of items related to these topics in the Civey item sample. To ensure the comprehensiveness and validity of our analysis, we considered items specifically related to either the COVID pandemic or the war in Ukraine. Further details are available in the S1 Appendix.

**Intercoder reliability.** To assess the intercoder reliability, we ran a Cohen's kappa test [50, 51]. Table 1 shows the kappa values for our independent variables.

Following Landis and Koch's [51] (p. 165) description of the relative strength of agreement associated with kappa (see Table 2), we obtained kappa values suggesting a substantial to almost perfect agreement.

## Results

### Descriptive statistics

Fig 2 depicts the overall polarization of the 817 items we analyzed. The distribution is unimodal and somewhat skewed to the right, indicating a tendency to answer the questions related to the items in a consensual way. Fig 2 highlights an important result for the opinion polarization debate: only about 20% of the items have a negative value on the Van der Eijk index, and negative values on the Van der Eijk index are all moderate (with -0,55 being the minimum value on a scale theoretically ranging to -1). In other words, only about 20% of our 817 attitudinal items on political issues show moderate opinion polarization. At the same time, we do not find any attitudinal items with a high level of opinion polarization.

We gain a more precise insight by subdividing items' value on the Van der Eijk index along their issue domains. Fig 3 illustrates the distribution of the Van der Eijk index along the items' issue domain. The issue domains are ordered from least to most polarizing. Descriptively, we see that items related to gender issues, as well as to education, research, and infrastructure issues are the items with the most agreement (see top of Fig 3). By contrast, items on environmental, Europe-related, and health issues are the most polarized (see bottom of Fig 3).

**Table 1. Intercoder reliability.**

| Variable | Kappa |
|---|---|
| Culture | 0.76 |
| Benefits | 0.67 |
| Minority | 0.70 |
| Abstraction | 0.84 |
| Technicality | 0.65 |

**Table 2. Kappa and strength of agreement.**

| Kappa | Strength of agreement |
|:---:|---|
| <0.00 | Poor |
| 0.00–0.20 | Slight |
| 0.21–0.40 | Fair |
| 0.41–0.60 | Moderate |
| 0.61–0.80 | Substantial |
| 0.81–1.00 | almost perfect |

To conclude the presentation of our dependent variable, Fig 4 brings some concrete insights into our sample of items with a list of items with the highest level of agreement, items with an exact uniform distribution, and items with the highest level of disagreement.

Turning now to our independent variables, Table 3 presents their descriptive statistics. On average, the issue domain of an item is mentioned over 8.700 times in newspapers the week before the survey data collection of the corresponding item. Out of the 817 items, every tenth relates to a cultural issue and roughly every fifth has a notion of costs included in its formulation. Items outlining a benefit are less frequent (14%). In 17% of the cases, an item addresses a minority-related issue. While a third of the items shows a low level of abstraction, only every tenth item is formulated in a highly abstract way. Similarly, also only about ten percent of items shows a high level of technicality, while every fourth item implies only little knowledge to answer the item. Finally, roughly every tenth item was COVID-, or Ukraine-related.

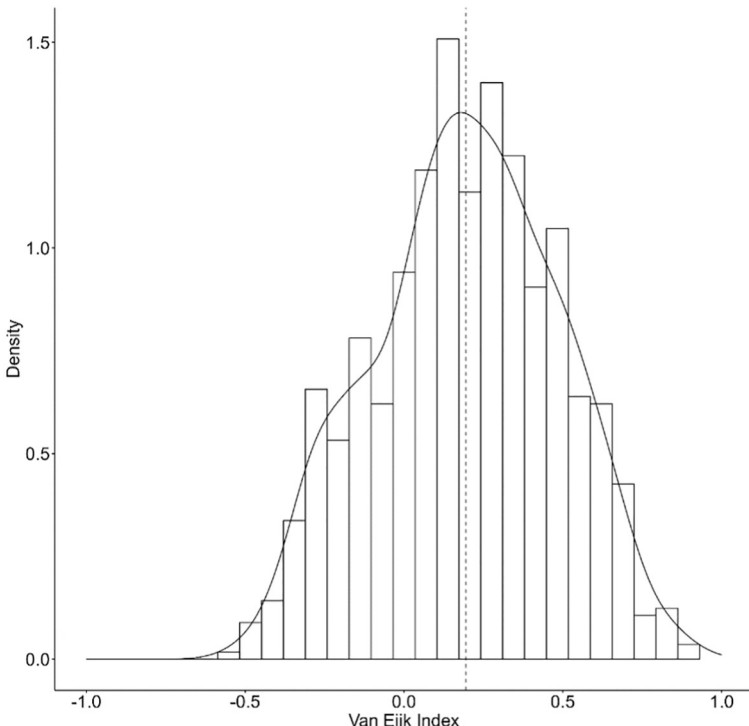

**Fig 2. Distribution of all analyzed items on the Van der Eijk index.**

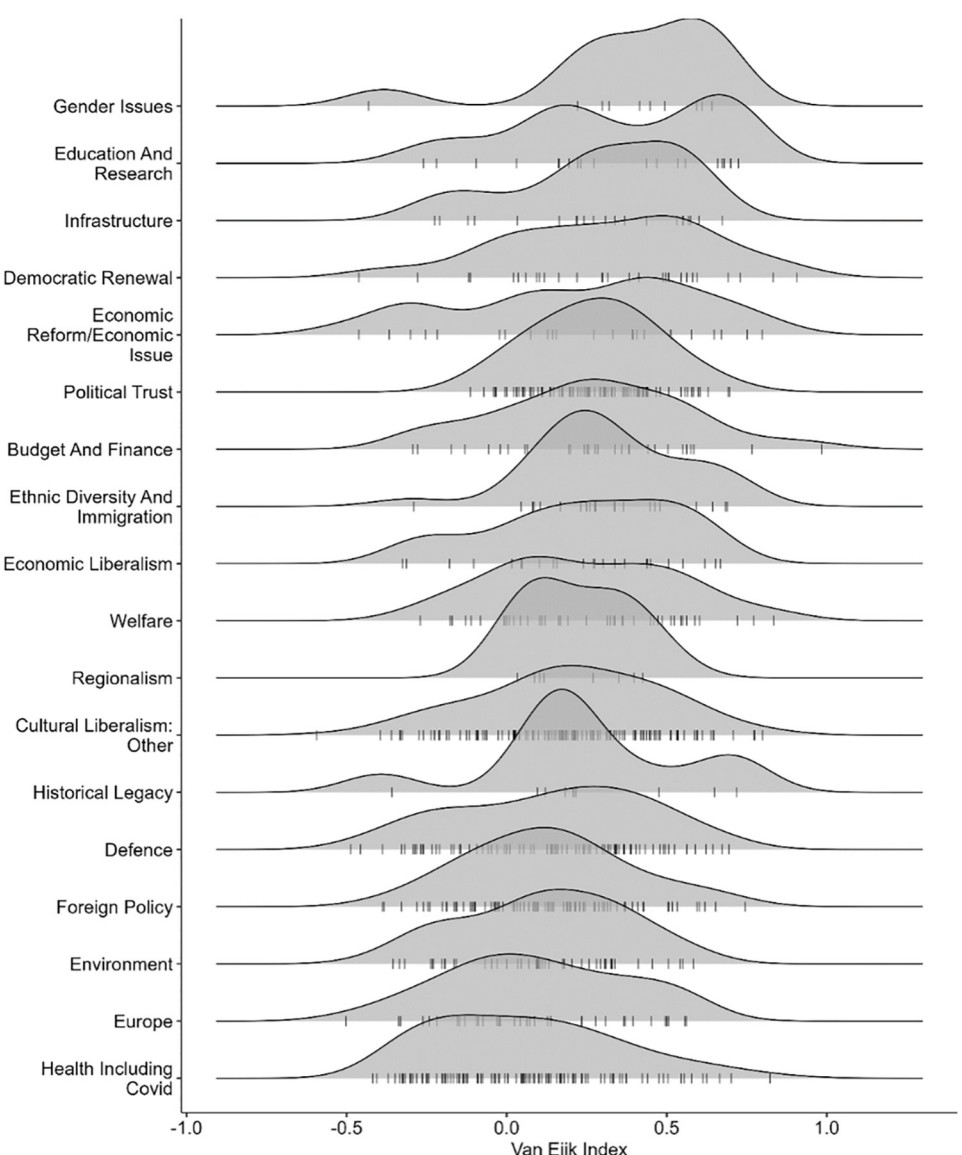

**Fig 3. Distribution of the items' issue domains on the Van der Eijk index.**

## Bivariate analysis of the Van der Eijk index

Fig 5 illustrates how the item characteristics impact distributive polarization by showing results of t-tests for nominal two-group comparisons (Hypothesis 2–4), variance analysis for multi-group comparisons (Hypotheses 5 and 6) and correlation for continuous variables (Hypothesis 1).

First, salience is significantly and negatively associated with the Van der Eijk index, meaning that the higher the media salience of an item's issue domain is, the more polarized an item is. This result is in line with our first hypothesis.

Second, an item belonging to the cultural issue domain positively affects the Van der Eijk index values (significant at the 5% level). In other words: if an item is culture-related,

| | Poll Text | Van der Eijk Index | Distribution |
|---|---|---|---|
| **High Agreement** | Should the federal government, in your opinion, take stronger action against the waste of tax money? | 0.95 | |
| | In your opinion, should electoral law be reformed to keep the number of members of parliament lower in the future? | 0.90 | |
| | How do you assess the extent of bureaucracy in Germany? | 0.87 | |
| | In your opinion, should there be exceptions for members of the Bundestag from generally applicable Corona rules? | 0.86 | |
| | Should people who have worked for a long time in their lives be entitled to a minimum pension? | 0.84 | |
| **Equal Distribution** | How do you rate the fact that Corona rules in schools are being lifted in almost all federal states after the Easter holidays? | 0.00 | |
| | How do you currently rate the work of Federal Minister of Labor Hubertus Heil on a scale of 1 (very good) to 6 (insufficient)? | 0.00 | |
| | Should the unemployment benefit II, known as 'Hartz IV', be retained in its current form or fundamentally changed? | 0.00 | |
| | How do you rate the performance of the traffic light cabinet in the federal government after about 100 days? | 0.00 | |
| | In your opinion, should Germany introduce an immediate ban on plastic bags? | 0.00 | |
| **High Polarization** | Should homosexual couples have the same adoption rights as heterosexual couples? | -0.45 | |
| | In your opinion, should Germany support Ukraine more with military equipment than it has so far? | -0.46 | |
| | What do you think of the EU Commission's plan to temporarily classify investments in nuclear energy as climate-friendly? | -0.48 | |
| | In your opinion, should the federal government maintain the current course in the Russia-Ukraine war, even if it has economic consequences for Germany and the population? | -0.51 | |
| | Should every adult be automatically registered as an organ donor, as long as they do not explicitly object? | -0.55 | |

**Fig 4. Top 5 items with the highest Van der Eijk value, top 5 items with a value closest to 0, and top 5 items with the lowest Van der Eijk value.**

**Table 3. Distribution of independent variables.**

| Variable | N | Mean | Min | Max |
|---|---|---|---|---|
| Salience | 817 | 8714.208 | 0 | 330183 |
| Cultural Issue | 817 | | | |
| . . . No | 750 | 91.8% | 0 | 1 |
| . . . Yes | 67 | 8.2% | 0 | 1 |
| Cost | 817 | | | |
| . . . No | 634 | 77.6% | 0 | 1 |
| . . . Yes | 183 | 22.4% | 0 | 1 |
| Benefits | 817 | | | |
| . . . No | 700 | 85.7% | 0 | 1 |
| . . . Yes | 117 | 14.3% | 0 | 1 |
| Minority Issue | 817 | | | |
| . . . No | 679 | 83.1% | 0 | 1 |
| . . . Yes | 138 | 16.9% | 0 | 1 |
| Level of Abstraction | 817 | | | |
| . . . Low | 272 | 33.3% | 0 | 1 |
| . . . Medium | 460 | 56.3% | 0 | 1 |
| . . . High | 85 | 10.4% | 0 | 1 |
| Level of Technicality | 817 | | | |
| . . . Low | 203 | 24.9% | 0 | 1 |
| . . . Medium | 533 | 65.2% | 0 | 1 |
| . . . High | 81 | 9.9% | 0 | 1 |
| COVID-related | 817 | | | |
| . . . No | 731 | 89.5% | 0 | 1 |
| . . . Yes | 86 | 10.5% | 0 | 1 |
| Ukraine-related | 817 | | | |
| . . . No | 716 | 87.6% | 0 | 1 |
| . . . Yes | 101 | 12.4% | 0 | 1 |

respondents are more likely to share consensual attitudes. This finding contradicts the postulated relationship we formulated in Hypothesis 2.

Third, items measuring attitudes toward financial and non-financial individual costs tend to have a significantly lower Van der Eijk index value than attitudinal items not assessing any costs. Furthermore, items measuring attitudes toward financial and non-financial individual benefits do not significantly differ in their Van der Eijk index value from attitudinal items not assessing any benefits. This finding is consistent with our Hypothesis 3: items measuring attitudes towards financial and non-financial individual costs polarize to a larger extent than items measuring attitudes toward financial and non-financial individual benefits.

By contrast, we don't find evidence for our fourth hypothesis stating that items targeting a minority group are more likely to polarize: the association between the level of polarization and the extent to which an item tackles a minority-related issue is not significant at conventional levels. Fifth, highly abstract items polarize respondents significantly less than items formulated more concretely, corresponding to our Hypothesis 5. By contrast, there is no significant difference between items with a low and a medium level of abstraction. Lastly, items requiring a medium level of technicality tend to be more polarized than items requiring a low level of technicality. This is in contrast with our postulated relationship in Hypothesis 6.

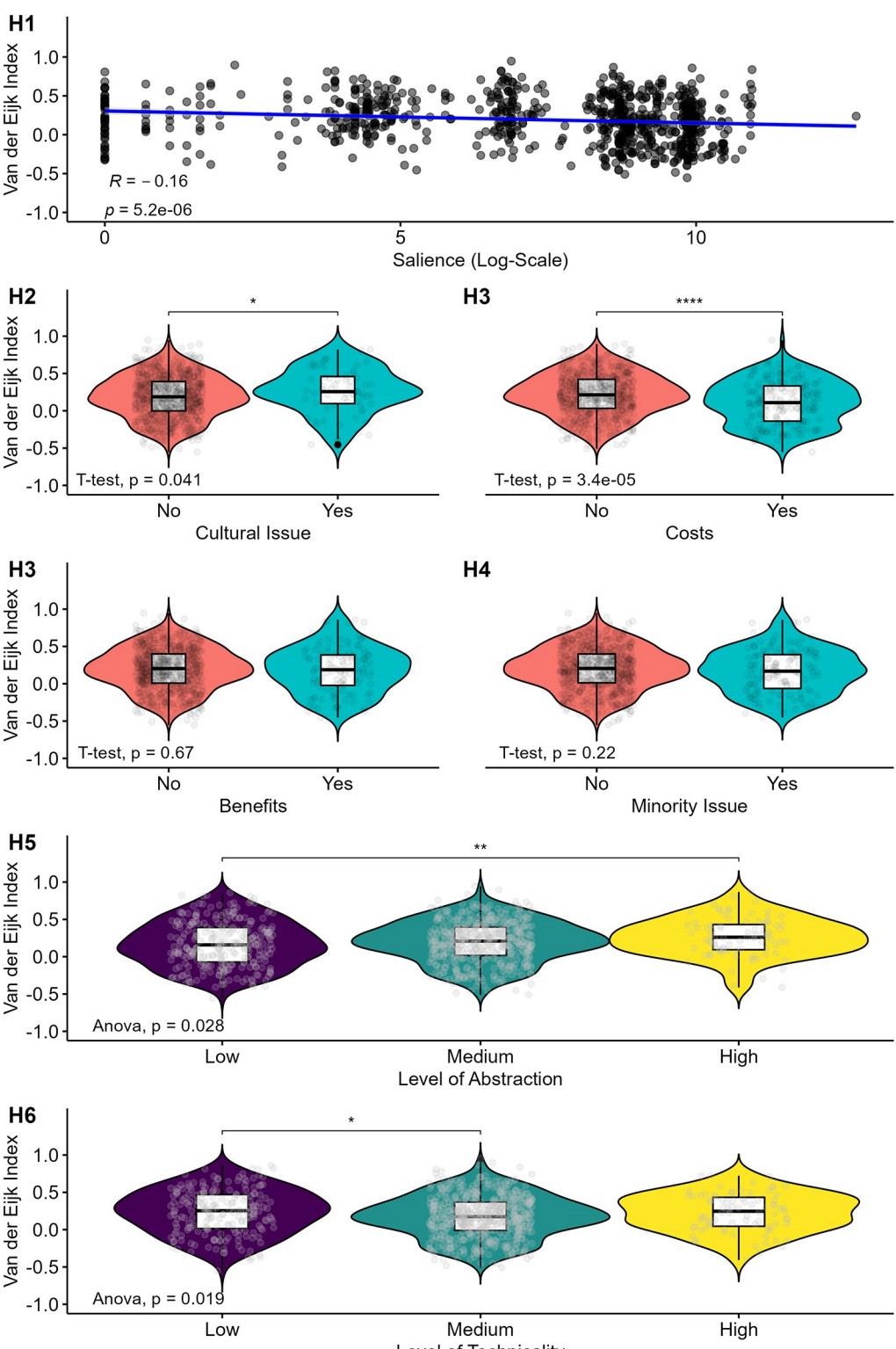

**Fig 5. Effects of item characteristics on the Van der Eijk index.** Stars denote the significance level of pairwise comparison: *p<0.05; **p<0.01; ***p<0.001.

There is no significant difference in the level of polarization between items with a high level of technicality and those with a low or medium level.

As a robustness check and to better understand how item characteristics affect the level of polarization, we employ ordinary linear least squares regressions to regress the Van der Eijk index on our independent variables (S1 Table in S1 Appendix). Model 1 presents a regression model with all our explanatory independent variables, while we included the control variables capturing whether an item is COVID- or Ukraine-related in the second model. Overall, we can confirm our bivariate findings.

## Conclusion

We investigated item characteristics that might influence distributive opinion polarization based on a unique sample of 817 items measuring political attitudes from a German non-random probabilistic online panel. We operationalized opinion polarization with the Van der Eijk agreement index that ranges from -1 (entire bimodal distribution) to +1 (entire unimodal distribution). Our study is explorative due to the lack of literature in survey methodology on this issue. We drew on insights from psychology, sociology, and political sciences to derive six hypotheses that might explain why survey item characteristics could lead to distributive opinion polarization.

A key finding to the opinion polarization debate is that only about 20% of the sampled items have a negative value on the Van der Eijk index and can thus be considered polarizing. Moreover, those polarizing items have only a moderate level of polarization. These results confirm the findings of previous studies on opinion polarization in Western Europe: only a minority of items polarize public opinion, and when items polarize, they do so only moderately. We hope that this important finding based on an extensive sample of survey items will help the scientific debate on opinion polarization to move beyond the mere question of the presence of distributive opinion polarization in Western Europe.

In addition to this relative lack of distributive opinion polarization, we could highlight the significant association of several item characteristics with the item level of polarization in our bivariate analyses.

First, items formulated abstractly are significantly related to a lower level of polarization. We interpreted this finding in light of the social psychological construal-level theory [37]. Accordingly, abstractly formulated items are more likely to activate broader commonly shared values and openness to other viewpoints and face, thus, a higher level of consensus. These abstractly formulated items tend to measure broader values or common goals (e.g., reducing social inequality, societal value of immigration, and gender equality). By contrast, concretely formulated items tend to focus on evaluating instruments to reach these common goals (such as policies). This finding is particularly important to understand the main results of previous studies on opinion polarization in Europe. As aforementioned, most previous studies measuring the presence of opinion polarization rely on national and international secondary social sciences survey data. Typical items from these social sciences surveys are formulated in a particularly abstract way (such as "The state should help reduce income inequality" or "Immigrants enrich cultural life"). According to our results, most of the previous studies focused on items that are formulated in such a way as to induce opinion agreement instead of opinion polarization. We therefore strongly recommend scholars working on opinion polarization with secondary social sciences survey data to reflect on the extent to which the formulation characteristics of the items analysed might affect their analysis and their potential implication on their empirical results.

Two further item characteristics were significantly associated with higher levels of polarization in accordance to our hypotheses: the item´s salience and items measuring costs. The fact that the salience of an item´s issue domain is significantly associated with a higher level of disagreement confirms the directly motivated reasoning theory [16]: mere exposure to information on an issue through the media leads significantly to more distributive opinion polarization. This finding highlights the importance of considering the relationship between issue salience and citizens' attitudes to enhance our understanding of opinion polarization. Furthermore, whether an item measures individual financial or non-financial costs is significantly related to a higher level of disagreement. According to prospect theory [27], costs–that tend to be perceived as losses related to the status quo–attract stronger and more affective reactions than benefits–which tend to be perceived in terms of gains. In the case of survey items, items measuring some individual financial or non-financial costs lead respondents to select more extreme answer categories, which implies a higher level of distributive polarization. Prospect theory seems to have some applicability in the survey methodology field.

However, three item characteristics were not significantly associated with the level of polarization in the expected way. First, we find that items with a medium level of technicality polarize more than items with a low level. This result is counterintuitive, given our theoretical argumentation derived from the concepts of hard issues [39] and political sophistication [40]. Our finding points indeed to the fact that items from our sample with a higher level of technicality do not seem to suffer from more measurement errors than those with a low level of technicality. According to our results, the discriminatory power of items with a high level of technicality is thus not lower than the discriminatory power of items with a lower level of technicality.

Second, and in contrast to what we were expecting from Norris and Inglehart's [24] backlash theory, items related to a cultural issue (i.e., gender, LGTBQI+, or ethnic minorities) receive a significantly higher level of agreement than items on non-cultural related issues. It should be remembered that our opinion polarization indicator (i.e., the Van der Eijk index) is only a measure of agreement and disagreement and does not say anything about the average position of respondents on an item. Thus, an item on a culture-related issue with a high level of agreement might imply a sizeable average opposition or a large average support. Third, we did not find evidence that items targeting a minority group are more likely to polarize. As our study has a pure explorative purpose, we lack theoretical foundations to provide any ad hoc explanations to these associations between item characteristics and level of distributive opinion polarization that contradict our expectations. These results might nevertheless provide some food for thought to the survey methodology debate.

While our study focused on survey items collected in a German non-probability online panel, we believe these findings will likely be generalizable to other contexts and periods. Indeed, while some items of our sample are particular to the German case (e.g., implementation of a speed limit on highways), the content of most of the 817 survey items is relevant to other countries and are likely to have been asked in a similar formulation in surveys in other countries. Moreover, the period in which the coded items were asked is characterized by two main issues (COVID pandemic and the war in Ukraine). Further robustness analyses show that our main findings remain stable even when controlling for these two major issues. Therefore, we are confident that our results will likely apply to other contexts and periods.

We would like to conclude our study by highlighting the likelihood of measurement error in the independent variables that were coded manually. The variables, whether an item measures individual costs or benefits, targets a minority group, is formulated abstractly or concretely, or encompasses a low or high level of technicality were coded manually by one of the authors. This coding exercise required a particularly intensive effort of interpretation and

judgement to provide a systematic and coherent coding frame. Indeed, many items differ from each other on these variables in a very granulated manner; the coder had thus to focus on semantic nuances in the coding effort, which is likely to induce measurement error. We can nevertheless be confident in these manually coded variables, as the intercoder reliability index pointed to a substantial agreement for all coded variables. However, compared to other text corpora that are coded manually, the coding effort for building our variables characterising survey items on political attitudes required decisions based on fine semantic nuances, which might have induced higher measurement error. Moreover, these manually coded variables are highly skewed: In our dummy variables, only between 8% (cultural issue variable) and 22% (cost-related variable) of the items exhibit the value 1. This means that the majority of data points do not fall under the definition underlying our variables. This, in turn, reduces the power of our bivariate analyses.

Despite these limitations of measurement error and statistical power related to most of our key independent variables, we could detect several significant bivariate associations between items´ characteristics and their polarizing power. This, in turn, highlights the potential explanatory power of our key independent variables in case of refinement of our analytical and measurement approaches. With this contribution, hopefully, we have opened a new research avenue at the crossroads of survey methodology and political sociology that will enable scholars to expand our understanding of distributive opinion polarization and the lack thereof. A further promising research avenue would be to assess the role of item formulation and item issue domains on other, more complex forms of polarization, such as group-based polarization [12] or affective polarization [13].

## Supporting information

**S1 Appendix.**
(DOCX)

## Acknowledgments

We thank the participants of the Colloquium organized by Johanna Gereke, Natascha Nisic, and Gunnar Otte at the University of Mainz, the participants of the OSI-Bag at the FU Berlin and Stephan Dochow-Sondershaus for their particularly creative comments on previous versions of our analyses. Furthermore, we thank Karl Schweizer for his helpful and reliable assistance. Lastly, we would like to thank Gefjon Off and Markus Wagner for their useful, insightful, and constructive comments as Reviewers of PlosOne on a previous version of this manuscript.

## Author Contributions

**Conceptualization:** Céline Teney.

**Data curation:** Céline Teney, Giuseppe Pietrantuono.

**Formal analysis:** Tobias Wolfram.

**Methodology:** Céline Teney, Giuseppe Pietrantuono, Tobias Wolfram.

**Project administration:** Giuseppe Pietrantuono.

**Supervision:** Céline Teney.

**Validation:** Giuseppe Pietrantuono.

**Visualization:** Tobias Wolfram.

**Writing – original draft:** Céline Teney.

**Writing – review & editing:** Giuseppe Pietrantuono, Tobias Wolfram.

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
