## [Decision Letter · Decision Letter 0]

23 Nov 2023

PONE-D-23-30566What polarizes citizens? An explorative analysis of 817 attitudinal items from a non-random online panel in GermanyPLOS ONE

Dear Dr. Teney,

Thank you for submitting your manuscript to PLOS ONE. After careful consideration, we feel that it has merit but does not fully meet PLOS ONE’s publication criteria as it currently stands. Therefore, we invite you to submit a revised version of the manuscript that addresses the points raised during the review process. Editors' comments: both reviewers provide several fair and reasonable comments that call for major and minor revisions of different kinds (e.g., theoretical, empirical). I believe that you should engage with all of the comments. I will send back the manuscript to both reviewers and invite them to review the revised iteration if you choose to re-submit it. 

We look forward to receiving your revised manuscript.

Kind regards,

Jean-François Daoust

Academic Editor

PLOS ONE

“This paper is part of a  research project on attitudinal polarization funded by the German Research Foundation (DFG).”

Reviewers' comments:

Reviewer's Responses to Questions

**Comments to the Author**

1. Is the manuscript technically sound, and do the data support the conclusions?

Reviewer #1: Yes

Reviewer #2: Yes

2. Has the statistical analysis been performed appropriately and rigorously? 

Reviewer #1: Yes

Reviewer #2: Yes

3. Have the authors made all data underlying the findings in their manuscript fully available?

Reviewer #1: Yes

Reviewer #2: Yes

4. Is the manuscript presented in an intelligible fashion and written in standard English?

Reviewer #1: Yes

Reviewer #2: Yes

5. Review Comments to the Author

Reviewer #1: The paper constitutes an interesting contribution to the study of opinion polarization. Given the field’s current focus on the US, the focus on Germany – a context marked by a multi-party-system with weaker partisan identification – constitutes an empirical contribution in itself. By testing the effect of survey item formulation on polarization, the study further brings an important methodological contribution to the field. Finally, testing competing and complementary theoretical explanations for polarization, it theoretically contributes to the field of polarization. Empirically, the study is based on an impressive data analysis and coding effort.

General comments:

- The research question(s): Are the authors really just testing whether the survey item formulation affects polarization? Isn’t the effect also due to the nature and substance of the issue that the items ask about? For instance, an issue’s salience or whether it is a cultural issue do not depend on the survey item formulation. In contrast, whether it is abstract or concrete can be a matter of item formulation. I would suggest that the authors more systematically distinguish between aspects related to issues’ substance and aspects related to survey item formulation.

- To what extent is the study specific to the German context or generalizable to other contexts?

- What is the advantage of including so many items? Couldn’t the analysis be done more efficiently?

On polarization:

- The authors define polarization as a bimodal distribution. However, there are other definitions/ aspects of definitions of polarization that the authors do not engage with (e.g. see Traber et al. 2022). What about other aspects of polarization such as sorting or whether there is some kind of group identity? It would be great to see some engagement with this literature and justification of the chosen definition and potential limitations of this definition.

- Further, I would like to challenge the authors on the account that opinion polarization is generally problematic. Is opinion polarization always considered problematic, or can it be an indicator of “healthy” pluralism, too? Does this perhaps depend on the issue, e.g. when an opposition to an issue is problematic for democratic principles? Sometimes, general (dis)agreement with an issue could also be problematic from a democratic perspective, couldn’t it?

On issue salience:

- The authors define issue salience as the coverage the news media affords a given issue. However, I would think that media coverage is a proxy/ measurement of issue salience rather than the definition of it. I would like to see an actual definition of what the authors mean by issue salience, and how media coverage captures it. Alternatively, the authors could specify that they only refer to salience in the media, rather than salience in general (see Wojcieszak et al, 2018).

- As regards the effects of issue salience, I don’t think that “echo chambers” are the only possible mechanism at play in explaining the role of issue salience in polarization. For instance, issue salience leads to more availability of information and exposure to information, which increases the likelihood of people taking more determined positions on the issue. The issue salience literature (see Dennison 2019) can help with elaborating on such mechanisms.

- Table 3: Could the authors show the minimum and maximum values of the salience variable?

On loss aversion:

- I wonder whether the authors could make use of the literature on material and symbolic threats with regard to theorizing the effects of (perceived) losses/ costs.

- Further, I wonder about the context- and perception-specific nature of losses and benefits. For instance, the example mentioned by the authors on the highway speed limit entails the cost that people aren’t allowed to speed on the highway, but it also entails a gain in road traffic safety. Similarly, the authors assess that lifting Covid test obligations is a benefit, however, this comes at the cost of a greater health risk. Whether an item is seen as entailing a cost or a benefit seems to depend a lot on individual perception. Such classification may thus be more ambiguous than proposed by the authors. I think it is safe to associate a cost with items specifically asking respondents about their willingness to pay for something, and to associate a benefit with tax incentives. However, I don’t think that other items are easily classified as entailing a cost or benefit. Therefore, I do not trust the current coding of the measurement and I would recommend the authors to apply a stricter definition of cost and benefit in more strictly financial terms.

On minorities:

- Which groups do the authors consider as minorities? E.g. are women considered as a minority? Some would say that they are because of their discrimination, others would say that they aren’t, because they are a large group in society. Similarly, not everyone would consider old people as a population group that is typically considered part of identity politics.

On abstract vs. concrete formulation:

- How do the authors deal with the fact that some concrete items require a lot of specific knowledge to make an assessment? For instance, the question “Should the federal government recognize Jerusalem as Israel's capital?” requires knowledge about the implications of such a recognition. Similarly, the question “How do you currently rate the work of Federal Minister of Labor Hubertus Heil on a scale of 1 (very good) to 6 (insufficient)?” requires knowledge about the work of Hubertus Heil. What does it mean to include items of which respondents are very unlikely to give informed answers, and for which a good level of knowledge is necessary? Could a lack of polarization on such items indicate that people just don’t know what to answer, rather than that they don’t have strong opinions on them? I would suggest removing items that require a high level of specific knowledge.

Literature recommendations:

- On polarization outside the US:

Traber, Denise, Stoetzer, Lukas F. and Burri, Tanja (2022) ‘Group-based public opinion polarisation in multi-party systems’, West European Politics, 46(4), pp. 652–677.

- On issue salience:

Dennison, James (2019) ‘A review of public issue salience: Concepts, determinants and effects on voting’, Political Studies Review, 17(4), pp. 436–446.

- On media coverage and polarization:

Wojcieszak, Magdalena, Azrout, Rachid and De Vreese, Claes (2018) ‘Waving the red cloth: Media coverage of a contentious issue triggers polarization’, Public Opinion Quarterly, 82(1), pp. 87–109.

- On why identity politics can be divisive:

Versteegen, Peter L. (2023) ‘The excluded ordinary? A theory of populist radical right supporters’ position in society’, European Journal of Social Psychology.

Reviewer #2: This paper examines attitude polarization in Germany, taking an approach focusing on survey methodology. The main question the paper seeks to answer is how many survey questions eliciting attitudes exhibit polarized response patterns. It then seeks to explain why some questions show more polarization than others, testing five hypotheses: salience, costs/benefits, minority group focus, culture, abstraction. Abstraction, salience and costs tend to have the strongest impact on attitude polarization.

This is generally a strong paper. I like the use of the public opinion data to get a large variation in attitude questions. I also find the results valuable in terms of getting researchers to think about how survey question formats influence findings concerning polarization, and how responses may vary in predictable ways.

That said, I have a few comments that mean some revisions are necessary.

- When it comes to issues, I was also wondering whether other divisions may be useful. One key distinction often made in the literature is between easy and hard issues (Carmines & Stimson 1980). This is not quite the same as the level of abstraction. The argument is that some topics are "easier" in that they require less complex answers - abortion or the death penalty are perhaps examples. The correct taxation policy is more of a hard issue. It surprised me that this common distinction was not considered or discussed. Similarly, I wondered about moralization as a related term that is used to distinguish different issues.

- The justification of the salience hypothesis is a bit odd. There is not a lot of evidence of echo chambers, at least online. Instead, in my view salience forces people to actually think about a topic and formulate their answer. Salience also means that elites have provided useful (often partisan) cues. These are stronger reasons why salience matters.

- The argument about non-financial costs and benefits is not clear. The example given makes it even less clear. How is a ban on imperial war flags a cost? It is not a cost for everyone. For many it would be a benefit! Maybe referencing clear policy proposals or policy change would be more useful.

- I am not sure minority targets should always lead to polarization, especially if the minority is small or strongly disliked.

- Can the authors write a little more about common rules of thumb for Kappa agreement scores? What is deemed a sufficient score? Do these scores account for how rare a category is? (It is easy to be very accurate if a feature is rarely present.)

- Table 1. What does Majority mean? Why is Culture missing?

Figure 3. The distributions should be bar charts, as only 5 answer categories existed.

Carmines, E. G., & Stimson, J. A. (1980). The two faces of issue voting. American Political Science Review, 74(1), 78-91.

6. PLOS authors have the option to publish the peer review history of their article (what does this mean?). If published, this will include your full peer review and any attached files.

Reviewer #1: No

Reviewer #2: **Yes: **Markus Wagner

---

## [Author Response · Author response to Decision Letter 0]

13 Feb 2024

Response Letter

What polarizes citizens? An explorative analysis of 817 attitudinal items from a non-random online panel in Germany

PONE-D-23-30566

This memo documents the changes we made to the paper in response to comments from the two reviewers. We want to thank the reviewers and the editor for the extremely helpful and constructive comments and the opportunity to send our revised paper for reconsideration. The comments were spot-on and helped us push the paper forward. The points raised by the reviewers are addressed one by one below (text in italics represents our answers to the concerns raised by the two referees):

Referee 1:

Referee 1’s comments are very helpful in revising the paper. We hope we can satisfactorily address the general remarks and the comments concerning the theory and the empirical analysis.

A. General comments:

First, the reviewer encouraged us to revise our research question and to more systematically distinguish between aspects related to issues’ substance and aspects related to survey item formulation. As she/they/he points out, we are not only testing whether the survey item formulation affects polarization but also whether the nature and substance of the issue that the items ask influences polarization. The reviewer states: “For instance, an issue’s salience or whether it is a cultural issue do not depend on the survey item formulation. In contrast, whether it is abstract, or concrete can be a matter of item formulation.”

Second, the reviewer raises the question of whether the study is specific to the German context or generalizable to other contexts.

Third, the reviewer asks what the advantage of including so many items is and if the analysis couldn’t be done more efficiently.

1. Research Question

We followed referee’s 1 suggestion and reframed our research questions. First, in the introduction, on p. 4, we now clarify our research questions: “We investigate the role of item characteristics in explaining the modality of the item response distribution. We take two characteristics into account: (a) the item’s issue (e.g., the nature or substance of an item), and (b) the item’s formulation.” 

Second, we reordered our hypothesis to match the separate research questions. Additionally, in the introduction, we altered a paragraph to read: “By drawing on the literature in psychology, sociology, and political sciences, we derive hypotheses on six item characteristics that we expect to influence distributive opinion polarization. The first two relate to the item’s issue, the last four to the way items are formulated: (i) the extent to which an item issue was salient in the news media before the survey data collection, (ii) the extent to which an item tackles a cultural issue, (iii) the extent to which an item involves individual costs or benefits, (iv) the extent to which an item targets a minorty group, (v) the level of abstraction of an item, and (vi) the level of technicality of an item.” Note that we also added a sixth hypothesis (see comments below in the section “F. On abstract vs. concrete formulation”). 

Third, we clarify again on p. 5 that we distinguish two sets of hypotheses: “While survey measurement research provides many general recommendations about survey item formulation (e.g., de Leeuw, Hox, and Dillman 2008; Schuman and Presser 1996), it does not specify the conditions under which a survey item is particularly likely to polarize respondents. We, therefore, draw on the literature in psychology, sociology, and political sciences to define two hypotheses on item issue characteristics and four hypotheses on item formulation characteristics and their role in distributive opinion polarization. In this section, we present our hypotheses on these item characteristics.”

Lastly, we changed the abstract accordingly: “Various studies point to the lack of evidence of distributive opinion polarization in Europe. As most studies analyse the same item batteries from international social surveys, this lack of polarization might be due to an item’s issue (e.g., the nature or substance of an item) or item formulation characteristics used to measure polarization. Based on a unique sample of 817 political attitudinal items asked in 2022 by respondents of a non-random online panel in Germany, we empirically assess the item characteristics most likely to lead to distributive opinion polarization – measured with the Van der Eijk agreement index. Our results show that only 20% of the items in our sample have some – at most moderate – level of opinion polarization. Moreover, an item’s salience in the news media before the survey data collection, whether an item measures attitudes toward individual financial and non-financial costs, and the implicit level of knowledge required to answer an item (level of technicality) are significantly associated with higher opinion polarization. By contrast, items measuring a cultural issue (such as issues on gender, LGTBQI+, and ethnic minorities) and items with a high level of abstraction are significantly associated with a lower level of polarization. Our study highlights the importance of reflecting on the potential influence of an item’s issue and item formulation characteristics on the empirical assessment of distributive opinion polarization.” 

2. Generalizability

The reviewer raises an important point regarding the generalizability of our study: Among the 817 items in our sample, only some of them are particular to the German context (e.g., the implementation of a speed limit on highways). Most of the other items are relevant to many other contexts. Moreover, we controlled in follow-up analyses for the two dominant issues covered by the sampled items (COVID pandemic and war in Ukraine) and could find similar results than the ones presented in the paper. We, therefore, are confident that our main findings apply also to other contexts and for other periods. We added a paragraph in the conclusion to clarify how our results may also hold in other contexts. The paragraph (p. 26) reads: “While our study focused on survey items collected in a German non-probability online panel, we believe these findings will likely be generalizable to other contexts and periods. Indeed, while some items of our sample are particular to the German case (e.g., implementation of a speed limit on highways), the content of most of the 817 survey items is relevant to other countries and are likely to have been asked in a similar formulation in surveys in other countries. Moreover, the period in which the coded items were asked is characterized by two main issues (COVID pandemic and the war in Ukraine). Further robustness analyses show that our main findings remain stable even when controlling for these two major issues. Therefore, we are confident that our results will likely apply to other contexts and periods.”

3. Inclusion of many items

The reviewer is totally right in pointing out that the inclusion of all the items comes at the cost of efficiency. However, we are convinced that from a substantial and methodological standpoint, the inclusion of a broad array of items provides a more comprehensive and detailed understanding of opinion polarization. We opted for the inclusion of all 817 items to allow for a more nuanced understanding of the phenomena. In our study, for instance, we find that only a minority (about 20%) of items tend to polarize public opinion. This result is significant as it challenges the general assumption of widespread opinion polarization and illustrates the importance of the analysis of an extensive sample.

B. On polarization

Reviewer 1 states that we “define polarization as a bimodal distribution. However, there are other definitions/ aspects of definitions of polarization that the authors do not engage with (e.g., see Traber et al. 2022). What about other aspects of polarization such as sorting or whether there is some kind of group identity? It would be great to see some engagement with this literature and justification of the chosen definition and potential limitations of this definition.”

We thank reviewer 1 for highlighting this missing argument in our paper. We revised the introduction and the conclusion, mentioning the restriction of our study to distributive opinion polarization, which can be considered the most straightforward and least complex form of polarization. In the introduction (p.4.), we added the following clarification: “Arguably, we focus in this study on the simplest form of opinion polarization measured with the modality of an item’s distribution among the overall sample of survey respondents, leaving aside more complex forms of polarization, such as group-based polarization (Traber, Stoetzer, and Burri 2022) or affective polarization (Wagner 2021). This focus on a single and simple form of opinion polarization is necessary to launch a scientific debate targeting the role of item formulation and the issue of items in opinion polarization.” In the conclusion, on p. 27, we added the sentence: “A further promising research avenue would be to assess the role of item formulation and item issue domains on other, more complex forms of polarization, such as group-based polarization (Traber, Stoetzer, and Burri 2022) or affective polarization (Wagner 2021).”

Further, reviewer 1 rightly challenges us on the account that opinion polarization is generally problematic. She/they/he writes: “Is opinion polarization always considered problematic, or can it be an indicator of “healthy” pluralism, too? Does this perhaps depend on the issue, e.g., when an opposition to an issue is problematic for democratic principles? Sometimes, general (dis)agreement with an issue could also be problematic from a democratic perspective, couldn’t it?”

To account for this comment, we rephrased the first sentence of the introduction (p. 3): “Opinion polarization has become a prevalent research and mediatic topic in Western Europe over recent years, mainly due to its scientific and societal relevance. Indeed, while multiple and cross-cutting opinion polarization might be conducive to social order in pluralist societies, opinion polarization combined with issue alignment can lead to political conflicts and thus threaten social cohesion and social order (DellaPosta and Macy 2015; DiMaggio, Evans, and Bryson 1996).” However, we refrained from discussing the normative debate on opinion polarization at length, as it would go beyond the purpose of our empirical study focusing on item formulation and item issues. 

C. On issue salience

Concerning issue salience, reviewer 1 raises two points: First, he writes: “The authors define issue salience as the coverage the news media affords a given issue. However, I would think that media coverage is a proxy/ measurement of issue salience rather than the definition of it. I would like to see an actual definition of what the authors mean by issue salience, and how media coverage captures it. Alternatively, the authors could specify that they only refer to salience in the media, rather than salience in general (see Wojcieszak et al., 2018).”

Second, the reviewer remarks: “As regards the effects of issue salience, I don’t think that ‘echo chambers’ are the only possible mechanism at play in explaining the role of issue salience in polarization. For instance, issue salience leads to more availability of information and exposure to information, which increases the likelihood of people taking more determined positions on the issue. The issue salience literature (see Dennison 2019) can help with elaborating on such mechanisms.”

We thank both reviewers for highlighting the limitations of the echo chamber theory for deriving our hypothesis on the role of media issue salience and distributive opinion polarization. In particular, we thank reviewer 1 for suggesting the work by Wojcieszak et al. (2018), which was indeed very relevant to us for rewriting the theoretical section for this hypothesis on media issue salience. We used a similar theoretical framework as the one proposed by Wojcieszak et al. (2018) to explain why mere exposure to issues through media is likely to lead to more radical opinion on an issue and, thus, to more distributive opinion polarization – based on the psychological theory of directly motived reasoning (Flynn, Nyhan, and Reifler 2017). Based on the reviewer’s suggestions, we rewrote the section on issue salience on pp. 5–6: “Our first hypothesis on item issue characteristics refers to the role of media issue salience in distributive opinion polarization. From the psychological theory of directly motivated reasoning (Flynn, Nyhan, and Reifler 2017), we can hypothesize that mere exposure to information on an issue through the media is likely to lead to distributive opinion polarization. Directly motivated reasoning refers to the (unconscious) strategy of people to seek out information that reinforces their preferences (i.e., confirmation bias), denigrate attitudinal incongruent arguments (i.e., disconfirmation bias), and evaluate information supporting their prior attitudes as stronger and more compelling than counter attitudinal information (i.e., prior attitude effect) (Taber and Lodge 2006, 757). Directional motivational reasoning implies that processing additional information on an issue is likely to sharpen citizens´ prior beliefs and attitudes on the particular issue, which in turn increases attitudinal polarization (Taber and Lodge 2006). Empirical studies have indeed shown that directly motivated reasoning leads citizens to endorse stronger opinions (i.e., be more polarized) on an issue after having been exposed to new information on this issue. This effect appears in particular among those who have strong prior opinion on the respective issue and those who are more politically knowledgeable, as the former have affective links to the issue and the latter possess more ammunition to counter information disconfirming their prior beliefs (Taber and Lodge 2006; Wojcieszak, Azrout, and De Vreese 2018).

The (unconscious) activation of directly motivated reasoning is independent of the content of the information to which one is exposed (i.e., whether the information content confirms or disconfirms prior beliefs) (Taber and Lodge 2006). Thus, the mere exposure to media news on an issue is likely to induce directly motivated reasoning among citizens (in particular, those with strong prior beliefs and those more politically knowledgeable) who would then hold more polarized opinions on the respective issue. Indeed, Wojcieszak et al. (2018) showed that citizens in the Netherlands who were both fervent supporters and opponents of the EU held more polarized opinions after being exposed to media news about the EU. Therefore, we expect items with a high issue salience in the media to be more polarizing. By media issue salience, we refer to the relative coverage the news media allocates to a given issue (Epstein and Segal 2000).”

Third, the reviewer asked that Table 3 show the minimum and maximum values of the salience variable.

We revised Table 4 (formally Table 3) following the reviewer’s suggestion (see pp. 21-22).

D. On loss aversion

Referee 1 raised to points concerning our cost/benefit variables. First: “I wonder whether the authors could make use of the literature on material and symbolic threats with regard to theorizing the effects of (perceived) losses/ costs.” Second: “Further, I wonder about the context- and perception-specific nature of losses and benefits. For instance, the example mentioned by the authors on the highway speed limit entails the cost that people aren’t allowed to speed on the highway, but it also entails a gain in road traffic safety. Similarly, the authors assess that lifting Covid test obligations is a benefit, however, this comes at the cost of a greater health risk. Whether an item is seen as entailing a cost or a benefit seems to depend a lot on individual perception. Such classification may thus be more ambiguous than proposed by the authors. I think it is safe to associate a cost with items specifically asking respondents about their willingness to pay for something, and to associate a benefit with tax incentives. However, I don’t think that other items are easily classified as entailing a cost or benefi

---

## [Decision Letter · Decision Letter 1]

4 Apr 2024

What polarizes citizens?

An explorative analysis of 817 attitudinal items from a non-random online panel in Germany

PONE-D-23-30566R1

Dear Dr. Teney,

We’re pleased to inform you that your manuscript has been judged scientifically suitable for publication and will be formally accepted for publication once it meets all outstanding technical requirements.

Kind regards,

Jean-François Daoust

Academic Editor

PLOS ONE

Additional Editor Comments (optional):

Reviewers' comments:

Reviewer's Responses to Questions

**Comments to the Author**

1. If the authors have adequately addressed your comments raised in a previous round of review and you feel that this manuscript is now acceptable for publication, you may indicate that here to bypass the “Comments to the Author” section, enter your conflict of interest statement in the “Confidential to Editor” section, and submit your "Accept" recommendation.

Reviewer #1: All comments have been addressed

Reviewer #2: All comments have been addressed

2. Is the manuscript technically sound, and do the data support the conclusions?

Reviewer #1: Yes

Reviewer #2: Yes

3. Has the statistical analysis been performed appropriately and rigorously? 

Reviewer #1: Yes

Reviewer #2: Yes

4. Have the authors made all data underlying the findings in their manuscript fully available?

Reviewer #1: Yes

Reviewer #2: Yes

5. Is the manuscript presented in an intelligible fashion and written in standard English?

Reviewer #1: Yes

Reviewer #2: Yes

6. Review Comments to the Author

Reviewer #1: (No Response)

Reviewer #2: I would like to thank the authors for their detailed and careful response to my suggestions. I am happy with the revisions made and now support publication.

7. PLOS authors have the option to publish the peer review history of their article (what does this mean?). If published, this will include your full peer review and any attached files.

Reviewer #1: **Yes: **Gefjon Off

Reviewer #2: **Yes: **Markus Wagner
